# Improving intelligent dasymetric mapping population density estimates at 30-meter resolution for the conterminous United States by excluding uninhabited areas

Jeremy Baynes[1], Anne Neale[1], Torrin Hultgren[2]

1. Center for Public Health and Environmental Assessment, US Environmental Protection Agency, Research Triangle Park, NC 27711, USA
2. EPA National Geospatial Support Team, ITS-EPA III Infrastructure Support and Application Hosting Contract, Research Triangle Park, NC 27711, USA

*Correspondence*: Jeremy Baynes (baynes.jeremy@epa.gov)

**Abstract.** Population change impacts almost every aspect of global change from land use, to greenhouse gas emissions, to biodiversity conservation, to the spread of disease. Data on spatial patterns of population density help us understand patterns and drivers of human settlement and can help us quantify the exposure we face to natural disasters, pollution, and infectious disease. Human populations are typically recorded by national or regional units that can vary in shape and size. Using these irregularly sized units and ancillary data related to population dynamics, we can produce high resolution, gridded estimates of population density through intelligent dasymetric mapping (IDM). The gridded population density provides a more detailed estimate of how the population is distributed within larger units. Furthermore, we can refine our estimates of population density by specifying uninhabited areas which have impacts on the analysis of population density such as our estimates of human exposure. In this study, we used various geospatial datasets to expand the existing specification of uninhabited areas within the United States (US) Environmental Protection Agency's (EPA) EnviroAtlas Dasymetric Population Map for conterminous United States (CONUS). When compared to the existing definition of uninhabited areas for the EnviroAtlas Dasymetric Population Map, we found that IDM's population estimates for U.S Census Bureau blocks improved across all states in CONUS. We found that IDM performed better in states with larger urban areas than in states that are sparsely populated. We also updated the existing EnviroAtlas Intelligent Dasymetric Mapping toolbox and expanded its capabilities to accept uninhabited areas. The updated 30 m population density for the CONUS is available via EPA's Environmental Dataset Gateway (Baynes et al., 2021 ); https://doi.org/10.23719/1522948) and EPA's EnviroAtlas (https://www.epa.gov/enviroatlas).

## 1. Introduction

Population density is a critical variable for understanding human-environment relationships. It has been recognized as an essential societal variable for studying human interactions with the environment and it is crucial for quantifying human exposure to natural hazards. Data on population density have facilitated global mapping of the changing human footprint on Earth's terrestrial surface (Venter et al., 2016). The drivers and patterns of human settlement and population growth are a key part of understanding this expanding human footprint. Population density data allow researchers to investigate the spatio-

temporal patterns of human settlement, monitor changes in those patterns, and investigate how urban areas expand (Fang et al., 2018;Wei et al., 2017;Fang and Jawitz, 2019;Taubenböck et al., 2019). Furthermore, population density maps have allowed researchers to identify natural drivers of population density such as elevation, temperature, and precipitation (Liu et al., 2019;Samson et al., 2011). Population density data offer insights about the impact of human settlement and the risks and exposure people face from the environment. Population density has been used to assess the impacts of human activity on coral reefs (Bellwood et al., 2012;Cinner et al., 2013;Morais et al., 2019). Considerable work has used population density data to quantify human exposure and vulnerability to natural disasters and pollution (Smith et al., 2019;Nicholls and Small, 2002;Carroll et al., 1997;Samoli et al., 2019;Nahayo et al., 2019;Nasiri et al., 2018;Yuan et al., 2019). For example, population data has been used to quantify U.S population exposure to fine particles as a part of reporting the costs and benefits of the Clean Air Act Amendments of 1990 (U.S. Environmental Protection Agency, 2011). In Vietnam, researchers identified critical values of population density where the risk of dengue fever is high (Schmidt et al., 2011). Globally, population density was found to be a significant driver of the origins of emerging infectious diseases from 1940 – 2004 (Jones et al., 2008).

In the United States (US), estimating population density usually involves distributing population counts collected within source units such as blocks, or block groups delineated by the U.S. Census Bureau. The Census Bureau, like many other organizations, relies on censuses and surveys to allocate people to source units. Population density is often simply estimated as the population count divided by the area for each source unit. However, the population recorded in these units can be disaggregated to provide estimates of how the population within source units is distributed. This disaggregation is important when source units are large, varying in shapes and sizes, or the population within the source units is not evenly distributed (Leyk et al., 2019). Various techniques have been used to allocate population counts from source units to estimate population density. Pycnophylactic interpolation estimates population density within source units using a grid of equal-sized cells (Tobler, 1979). The pycnophylactic property of this method ensures that the counts from each source unit are maintained in the process and that population is not lost nor displaced beyond the source unit within which it was recorded (Tobler, 1979). Source units can be divided up into smaller target units of homogenous population density. For example, target units can be determined by the spatial intersection between census blocks and land cover classes. In this example, a target unit consists of the area of a land cover class inside a census block. Areal weighting distributes the population of source units to target units by the proportion of the area of the target unit inside the source unit (Goodchild and Lam, 1980). This method maintains the counts of the source units as suggested by Tobler (1979). However, the only determinant of population density is the area of a target unit inside a source unit. This is problematic where area might not be the best indicator of population dynamics. For example, in a source unit that is largely covered by a wildlife refuge and minimally covered by urban land use, the proportion of the source unit's population that resides in urban land use should, in reality, be greater than that in the wildlife refuge.

Dasymetric allocation of population can incorporate the population dynamics that are to be expected within source units in order to estimate population density. Dobson et al. (2000) used coefficients calculated by weighted combinations of factors

that influence human populations to estimate population density from aggregate population counts. Other methods have used

the Random Forest algorithm to predict population density at fine scales using aggregate population counts and aggregated fine-scale covariates that are related to population density (Sorichetta et al., 2015;Stevens et al., 2015). Researchers have modeled gridded population density from small area sampling of population counts rather than using a national census (Weber et al., 2018). To improve estimates, various dasymetric population mapping methods have used land use/land cover, climatic and topographic variables such as temperature, precipitation, elevation, and slope, and socio-economic variables such as

nighttime lights, roads, and points of interest related to human activity (Karunarathne and Lee, 2019;Lloyd et al., 2019;Ye et al., 2019). Dmowska and Stepinski (2017) used dasymetric modeling with a hybrid land cover and land use map to produce a U.S.-wide grid of population density at 30 m resolution. Their effort estimated land cover densities using nation-wide sampling of homogeneous Census Blocks but left open the possibility that local sampling from smaller spatial extents could improve results. Mennis and Hultgren (2006) developed an Intelligent Dasymetric Mapping (IDM) technique that estimates population

density by determining class-specific representative population densities from an ancillary raster containing classes that are indicative of population dynamics. IDM relies on a limited number of required input datasets, an ancillary raster and population source units. This makes IDM an appealing method over other promising, but more complex, methods (e.g., machine learning) because of its usability among broad audiences and applicability at various locations and scales. In 2016, IDM was used to develop a dasymetric population map of the conterminous US (CONUS) by the Environmental Protection Agency's (EPA)

Office of Research and Development. The map was developed for EnviroAtlas, an online collection of interactive tools and resources that provides data, research, and analysis on the relationships between nature, people, health, and the economy (Pickard et al., 2015). Census block counts for 2010 were disaggregated to 30 m grid cells using the 2011 National Land Cover Database (NLCD) as the ancillary raster. The identification of uninhabited areas and not allocating people to those areas can further refine population density to areas where humans are more likely to settle. This refinement has a marked impact on the

accuracy of estimates of population density (Fang and Jawitz, 2018;Smith et al., 2019;Leyk et al., 2019).

Uninhabited areas in the 2016 EnviroAtlas dasymetric population map effort were identified as the open water, perennial ice/snow, and emergent herbaceous wetlands land cover classes along with areas that have a slope greater than 25%. In this study, we updated the pre-existing EnviroAtlas dasymetric population map for the CONUS by incorporating additional geospatial data sets to expand areas identified as uninhabited. We then conducted an assessment to test the validity of our

methods and measure any improvement in population density mapping associated with our effort. While updating the EnviroAtlas dasymetric population map, we also updated the EnviroAtlas IDM toolbox, a toolbox originally developed for ESRI ArcMap 10.3 that allows users to create dasymetric population maps of their own study areas. The updated methodology has been implemented as a toolbox for ArcGIS Pro and a standalone Python tool that relies on open source libraries. We expanded the IDM toolbox's capabilities to accept additional uninhabited areas from users.


## 2. Data

We updated the existing population density map for CONUS using data that were nationally consistent and complete, fit for purpose, freely available or available under existing license, and relevant to human land use. Table 1 presents the data sets and layers that were used to update the dasymetric population map.


**Table 1. Datasets used for updating the EnviroAtlas dasymetric population map. IDM uses in bold were used in the 2016 EnviroAtlas dasymetric population map. (m/u: possible mixed-use feature)**

| Source | Dataset / Version / Format / Data Type | Data Name | IDM use |
|---|---|---|---|
| U.S. Census Bureau | Census blocks Vintage, 2010 TIGER/Line ESRI Shapefile Vector - Polygon | Census blocks with population and housing counts | **Source units** |
| Multi-Resolution Land Characteristics Consortium/ National Land Cover Database | 2011 Land Cover Version 2 (2016) ERDAS Imagine Raster - 30 m | Developed, Open Space | **Ancillary Class** |
| | | Developed, Low Intensity | **Ancillary Class** |
| | | Developed, Medium Intensity | **Ancillary Class** |
| | | Developed, High Intensity | **Ancillary Class** |
| | | Barren Land (Rock/Sand/Clay) | **Ancillary Class** |
| | | Evergreen Forest | **Ancillary Class** |
| | | Mixed Forest | **Ancillary Class** |
| | | Shrub/Scrub | **Ancillary Class** |
| | | Grassland/Herbaceous | **Ancillary Class** |
| | | Pasture/Hay | **Ancillary Class** |
| | | Cultivated Crops | **Ancillary Class** |
| | | Woody Wetlands | **Ancillary Class** |
| | | Emergent Herbaceous Wetlands | **Ancillary Class** |
| | | Perennial Ice/Snow | **Ancillary Class** |
| | | Open Water | **Ancillary Class** |
| | Developed Imperviousness Descriptor 2016 Edition, 2011 ERDAS Imagine Raster - 30 m | Primary road in urban area | Uninhabited Area |
| | | Primary road outside urban area | Uninhabited Area |
| | | Energy production site in urban area | Uninhabited Area |
| | | Energy production site outside urban area | Uninhabited Area |
| HERE/ NAVSTREETS | Land Use A 9,0, 2017 ESRI Geodatabase Vector - Polygon | Shopping center | Uninhabited Feature (m/u) |
| | | Industrial complex | Uninhabited Feature (m/u) |
| | | Cemetery | Uninhabited Feature |
| | Land Use B 9,0, 2017 ESRI Geodatabase Vector - Polygon | Aircraft roads | Uninhabited Feature |
| | | Retail | Uninhabited Feature (m/u) |

| | | | |
|---|---|---|---|
| OpenStreetMap Foundation (OSMF) & Contributors | Land use 2019 ESRI Shapefile Vector - Polygon | Commercial | Uninhabited Feature (m/u) |
| | | Mall | Uninhabited Feature (m/u) |
| | | Industrial | Uninhabited Feature (m/u) |
| | Places of interest 2019 ESRI Shapefile Vector - Polygon | Supermarket | Uninhabited Feature (m/u) |
| | | School | Uninhabited Feature |
| North American Rail Network | Rail network 2019 ESRI Shapefile Vector - Line | Rail network | Uninhabited Feature |
| CoreLogic | Residential parcels 2018 ESRI Geodatabase Vector - Polygon | Residential parcels | Inhabited Feature |
| U.S. Geological Survey Gap Analysis Project/ Protected Areas Database of the U.S. | Combined Protected Areas: Proclamation, marine, fee, designation, easement 2.0, 2018 ESRI Geodatabase Vector - Polygon | Local park | Uninhabited Feature |
| | | State park | Uninhabited Feature |
| | | State forest | Uninhabited Feature |
| | | National wildlife refuge | Uninhabited Feature |
| | | National forest | Uninhabited Feature |
| | | National park | Uninhabited Feature |
| | | National lakeshore/seashore | Uninhabited Feature |
| | | National grassland | Uninhabited Feature |
| U.S. Geological Survey | 2012 Raster - 30 m (projected to match NLCD) | National Elevation Dataset | **> 25% slope = Uninhabited Area** |

## 2.1 Boundaries

The TIGER/Line shapefiles from the United States Census Bureau provided state boundaries along with their Federal Information Processing Series (FIPS) codes (U.S. Census Bureau, 2012). The boundaries for statistical entities from the U.S. Census Bureau are organized hierarchically from census blocks within block groups which are contained within census tracts within the counties of a state (U.S. Census Bureau, 2012). We used a special release shapefile of the 2010 TIGER/Line census blocks that included the population and housing counts from the 2010 decennial census carried out by the U. S. Census Bureau

(U.S. Census Bureau, 2012). The shapefile also includes the state FIPS code, county FIPS code, the census tract code, and the census tabulation block number for each block (U.S. Census Bureau, 2012).

## 2.2 Land Cover

The 30 m, 2011 land cover classification from the 2016 NLCD (i.e., NLCD2016 2011) was used as the ancillary raster (Yang et al., 2018;Homer et al., 2020). Yang et al. used a leaf-on Landsat image as the base image for the 2011 NLCD classification.

Pixels with cloud, shade, and other anomalies in the base Landsat image were filled using leaf-on or leaf-off Landsat images

within two years of the base image (Yang et al., 2018). The NLCD classification was carried out using a decision-tree classifier with the Landsat image and ancillary data (Yang et al., 2018). The overall users accuracy for NLCD2016 2011 is 86.8% (Wickham et al., 2021)

## 2.3 Land Use

In order to identify uninhabited areas, we used several publicly available and proprietary datasets from the OpenStreetMap Foundation & Contributors (OSM), NAVSTREETS, CoreLogic, the Protected Areas Database of the U.S. (PAD-US), the North American Rail Network (NARN), NLCD, and the National Elevation Dataset (NED)(U.S. Geological Survey;OpenStreetMap contributors, 2019;CoreLogic, 2018;HERE, 2017;U.S. Geological Survey, 1999;Yang et al., 2018). From these data, we used several vector features and rasters related to built structures, zoning, topography, and protected areas.

Volunteers contribute and maintain geospatial data about roads, rail roads, built structures, land use, parks, and various other categories for OSM (OpenStreetMap contributors, 2019). NAVSTREETS provides boundaries for built structures and land use and CoreLogic provides boundaries for residential and non-residential parcels (CoreLogic, 2018;HERE, 2017). PAD-US is produced by the United States Geological Survey (USGS) Gap Analysis Program and provides nation-wide spatial data outlining the boundaries of protected open space held by national, state, and regional/local governments, and non-profit

conservation organizations (U.S. Geological Survey, 2018;Gergely and McKerrow, 2016). NARN is managed by the Federal Railroad Administration and is a comprehensive database of the US railway system (Federal Railroad Administration, 2019). NLCD includes a Developed Impervious Descriptor product that classifies the NLCD's percent impervious product into types of roads and energy production (Yang et al., 2018). The impervious product was developed by MRLC using regression tree models with Landsat imagery and training datasets generated from nighttime lights imagery (Yang et al., 2018).

## 3 Methods

### 3.1 Uninhabited features

Uninhabited features were identified and prepared for each CONUS state and Washington D.C. The goal of this step was to produce a single layer of uninhabited features for each state that would be used to reclassify NLCD pixels to a new uninhabited land cover class. From NAVSTREETS, we identified shopping centers, industrial complexes, cemeteries, aircraft roads, and

rail roads as uninhabited. A 30 m buffer was created around aircraft road centerlines and a 15 m buffer was created around railroad centerlines to ensure that all line features were converted to raster. Because we could find no existing railyard polygon data, railyard polygons were derived from railroad lines in NARN. We approximated railyard extents by applying a 500 m buffer around all rail line features with "YARDS' in the name field and then dissolving the resulting polygons into one feature. We then applied a negative 480 m buffer to the results of the 500 m buffer to ensure we were not capturing areas outside the

extent of the rail lines. These areas were identified as uninhabited. From OSM we identified retail, commercial land use, malls, industrial complexes, supermarkets, and schools as uninhabited (Table 1). Additionally, we designated local parks, state parks, state forests, national wildlife refuges, national forests, national parks, national lake shore or seashore, and national grasslands from PAD-US as uninhabited (Table 1).

The possibility of housing within the areas we identified as uninhabited warranted additional attention before marking the
entire area as uninhabited. For example, national forests have experienced an estimated housing growth of about 940,000 units
between 1940 and 2000 within their boundaries (Radeloff et al., 2010). In order to allocate potential population within areas
identified as uninhabited, we removed (i.e., spatially clipped) areas covered by residential parcels within all uninhabited
features listed in Table 1. We used the residential parcels from the area parcel feature class from CoreLogic (2018). Residential
parcels in this dataset included typical single-family residences; however, multi-family dwellings including apartment
complexes, urban mixed-use, and retirement communities were often considered commercial properties. We found no
consistent method to isolate these multi-family inhabited land-use parcels from other uninhabited commercial parcels;
therefore, we could not identify all commercial parcels as uninhabited.

Mixed-use zones may contain census blocks with a mix of retail, commercial, civic, business, industrial, and residential land
uses (Moos et al., 2018;Song and Knaap, 2004). Several of the land use types we identified as uninhabited can exist in mixed
use zoning and thus potentially be inhabited. From OSM and NAVSTREETS, we labeled shopping centers, industrial
complexes, malls, and supermarkets along with retail and commercial land uses as areas we initially identified as uninhabited
that can be found in mixed-use zoning (Table 1). If the combined area of these features covered greater than 90% of the entire
census block area, that block was labeled as mixed-use and those features within that block were excluded from our uninhabited
features.

Furthermore, if the combined area of features we identified as uninhabited covered more than 99% of a census block, all
features within that block were excluded from our uninhabited features. This way, if a census block was covered almost entirely
by uninhabited features, any population recorded in that block would not be lost. Uninhabited vector features remaining after
excluding residential parcels, mixed-use features, and features that covered more than 99% of a block were projected to Albers
Conical Equal Area projection and used as the uninhabited features for IDM (Fig 1a). The updated IDM toolbox reclassifies
ancillary raster pixels that coincide with uninhabited features to a new uninhabited ancillary class.

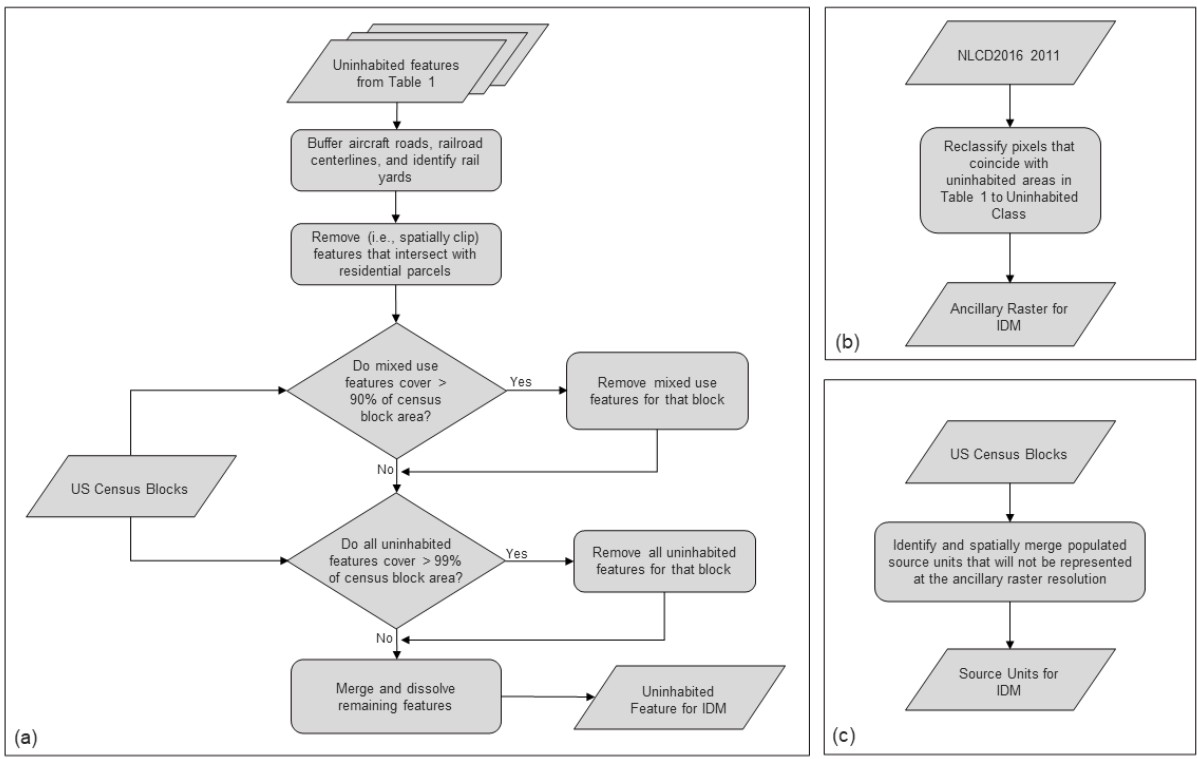

Figure 1. Data preparation workflow for uninhabited features (a), ancillary raster (b), and source units (c) for IDM processing.

### 3.2 Ancillary Raster

NLCD2016 2011 was the basis for the ancillary raster. We retained the only non-land cover attribute for identifying
uninhabited areas from the 2016 EnviroAtlas dasymetric population map; areas with a slope of greater than 25 % were
considered uninhabited. The percent slope was calculated from the National Elevation Dataset using GDAL (GDAL/OGR
contributors, 2019). In addition to slope, we used other gridded datasets to mask uninhabited areas. Land cover pixels that
coincided with uninhabited area pixels from Table 1 (i.e., primary roads and energy production classes from the Developed
Imperviousness Descriptor, or a slope of greater than 25%) were reclassified to a new uninhabited land cover class. This
reclassified NLCD classification was used as the ancillary raster for IDM (Fig 1b).

### 3.3 Source Units

The U.S. Census Bureau blocks with associated population counts from the 2010 decennial census were used as source units
for IDM. The IDM toolbox converts source units into a raster that matches the spatial resolution and extent of the input ancillary
dataset. Small or irregularly shaped source units that do not coincide with the center of a pixel at the ancillary dataset resolution
will not be represented in the derived raster and the population in that unit will not be included in the estimate of population

density. To account for the population in these blocks, we identified any populated census block that would not be represented in a 30 x 30 m pixel. These blocks were spatially merged and had their population added to the neighbouring block that met all the following criteria:

1. had the longest shared border.
2. was in the same census tract.
3. had a population greater than zero.

If no neighbouring block in the same census tract had population, then criteria 3 was dropped. This allowed us to account for population in these small blocks while not displacing the population outside of the census tract and limiting displacing population into unpopulated blocks. Census blocks were projected to Albers Conical Equal Area projection to match the NLCD. This modification of the 2010 census blocks was used as the source units for IDM (Fig 1c).

## 3.4 Intelligent Dasymetric Mapping

The IDM method from Mennis and Hultgren (2006) was used to estimate the population density (people per pixel). The modified 2010 U.S. Census Bureau blocks with associated population were used as source units and the NLCD reclassified to incorporate uninhabited areas was used as the ancillary raster. The target units were created by the spatial intersection between NLCD classes and U.S. Census Bureau blocks. Therefore, each target unit consists of the area of an NLCD class inside a block. A homogenous gridded population density (30 x 30 m) was estimated for each target unit inside the census blocks.

In order to estimate the population density for the target units, a representative population density was estimated for each land cover class from NLCD for each state. The representative population density of a land cover class is the number of people per pixel that were expected to reside in that land cover class throughout the state. IDM offers three ways to estimate the representative population density for an ancillary class. First, a representative population density can be set for an ancillary class from expert or domain knowledge or previous research. In line with the 2016 specification of uninhabited areas, the representative population density for the following land cover classes from NLCD was preset to zero people/pixel: open water, perennial ice/snow, and emergent herbaceous wetlands. Since we added an additional "uninhabited" class to the NLCD classification, we also set the preset density for this class to zero people/pixel. Second, the representative population density for an ancillary class can be sampled from source units that are considered representative of that ancillary class. The IDM toolboxes we developed allow users to set sampling eligibility thresholds. For this effort we determined that a representative block, $b$, for a sampled land cover class, $s$, met the following criteria:

1. Ninety five percent of the area of the source unit $b$ was covered by land cover class $s$.
2. The area of source unit $b$ was greater than 900 m$^2$ (1 pixel).

At least three representative census blocks were required for a land cover class to be considered sampled. After collecting all the representative blocks for a sampled land cover class, the representative population density for the class was estimated as (Mennis and Hultgren, 2006):

$$\widehat{D_s} = \sum_{b=1}^{m} y_b / \sum_{b=1}^{m} A_b \tag{1}$$

where:

$\widehat{D_s}$ = the representative population density of sampled land cover class $s$

$y_b$ = the population count of census block $b$

$A_b$ = the area of census block $b$

$m$ = the number of representative blocks for class $s$

Since the entire area of the block is used to distribute population counts in Eq. (1), only using blocks where 95% of the area is covered by the sampled land cover class ensures that the representative population density estimated for the class is based on 225   homogenous blocks. Lastly, intelligent areal weighting (IAW) was used to calculate the representative population density for all land cover classes within each state where insufficient representative blocks were found and no representative population density was preset. By this point, a representative population density had been determined by either a sampled or preset representative population density for land cover class $k$ (i.e., $\{k \in C \mid k \in (P \cup S)\}$ where $C$ is the set of all ancillary classes, $P$ is the set of all preset ancillary classes, and $S$ is the set of all sampled ancillary classes). IAW calculates the remaining 230   population counts for each source unit after sampled and preset representative population densities have determined a population estimate for target units in the source unit when possible (Mennis and Hultgren, 2006):

$$G_b = y_b - \sum_{t(k) \in b} \widehat{D_k} A_{t(k)} \tag{2}$$

where:

$G_b$ = the remaining census population count for block $b$

$\widehat{D_k}$ = the representative population density of land cover class $k$

$A_{t(k)}$ = the area of the target unit associated with land cover class $k$ in census block $b$

After calculating the remaining population for each block, an initial population was allocated to a given block's target units associated with land cover class $i$ that had not been determined by either a sampled or preset representative population density (i.e., $\{ i \in C \mid i \notin (P \cup S)\}$). IAW uses areal weighting to distribute the remaining census counts to the remaining target units (Mennis and Hultgren, 2006):

$$\hat{y}_{t(i)} = \begin{cases} 0, if \ G_b < 0 \\ G_b(A_{t(i)}/ \displaystyle\sum_{t(i)\in b} A_{t(i)}), if \ G_b \geq 0 \end{cases} \qquad (3)$$

where:

$\hat{y}_{t(i)}$ = the initial estimated population count for the target unit associated with land cover class $i$ in block $b$

$A_{t(i)}$ = the area of the target unit associated with land cover class $i$ in block $b$

Equation (3) differs slightly from the methods of Mennis and Hultgren in that here an initial population of zero was allocated to unsampled land cover classes if the total population estimated for sampled or preset classes in the block exceeded the census

count for the block. Although not explicitly stated in Mennis and Hultgren, this was implied as it avoids negative population estimates attributed to target units. After the initial population counts were estimated for each target unit associated with land cover class $i$, the representative population density of land cover class $i$ was determined as (Mennis and Hultgren, 2006):

$$\hat{D}_\iota = \sum_{t(i)=1}^{p} \hat{y}_{t(i)} \ / \sum_{t(i)=1}^{p} A_{t(i)} \qquad (4)$$

where:

$\hat{D}_\iota$ = the representative population density of land cover class $i$

$p$ = the number of target units in the study area that are associated with land cover class $i$

After the representative population density for each land cover class was determined using either a preset density, sampling (Eq. (1)), or IAW (Eq. (4)), the final population estimate for target unit $t$ which consists of the area of a land cover class $c$ (i.e., $\{ c \in C \}$) inside block $b$ was calculated as (Mennis and Hultgren, 2006):

$$
\hat{y}_t = \begin{cases} y_b \left( A_t / \sum_{t=1}^{n} A_t \right), & if \ \sum_{t=1}^{n} \widehat{D_{c(t)}} = 0 \\ y_b \left( A_t \widehat{D_{c(t)}} / \sum_{t=1}^{n} A_t \widehat{D_{c(t)}} \right), & if \ \sum_{t=1}^{n} \widehat{D_{c(t)}} > 0 \end{cases}
\tag{5}
$$

where:

$\hat{y}_t$ = the population estimated for target unit $t$ associated with land cover class $c$ in block $b$

$n$ = the number of target units in block $b$

$A_t$ = the area of target unit $t$

$\widehat{D_{c(t)}}$ = the representative population density of land cover class $c$ associated with target unit $t$

Equation (5) ensured that the population was not displaced beyond the block (Mennis and Hultgren, 2006). Equation (5) is also a slight deviation from Mennis and Hultgren in that area weighting would be used for population within a block made up entirely of land cover classes with representative population densities estimated at or preset to zero. Although rare, there were instances of populated Census blocks composed entirely of these land cover classes. This modification ensured any population within these blocks were not lost without giving weight to any specific land cover class. The final population density for a target unit $t$ that is associated with ancillary class $c$ and source unit $b$ can be calculated as (Mennis and Hultgren, 2006):

$$
\widehat{d_t} = \hat{y}_t / A_t
\tag{6}
$$

where:

$\widehat{d_t}$ = the population density (people / pixel) estimated for target unit $t$

We chose to apply IDM using sub-national zones versus a national analysis. States were selected as zones because they are generally large enough to collect a suitable number of homogenous source units for sampling while being small enough to represent some of the heterogeneity in population density across the CONUS. The input blocks, uninhabited features, and land cover rasters were prepared for each CONUS state and Washington D.C. In order to increase the number of representative blocks, all data for Rhode Island were combined with neighbouring Massachusetts. Likewise, data for Washington D.C. were combined with Maryland. Representative population densities were determined for 17 land cover classes in 47 'states' in the US for a total of 799 estimated densities (Fig. 2). Four land cover types were preset at zero for every state. Of the 611 unique

land cover type / state combinations that were not initially preset at zero, 596 were determined with sampling, 14 were determined using IAW, and one was preset (Fig. 2). In Connecticut, the representative population density for scrub/shrub was estimated at 3.4 using IAW. This would have resulted in shrub/scrub having the highest representative population density for any land cover type in the state and the estimate was over six standard deviations above the mean for that land cover type in all states. We chose to rerun IDM for Connecticut using the average representative density for scrub/shrub from all other states as a preset density. Population density was determined for each NLCD pixel within each state then joined to create a seamless 30 m population density estimate for the CONUS (Fig. 3).

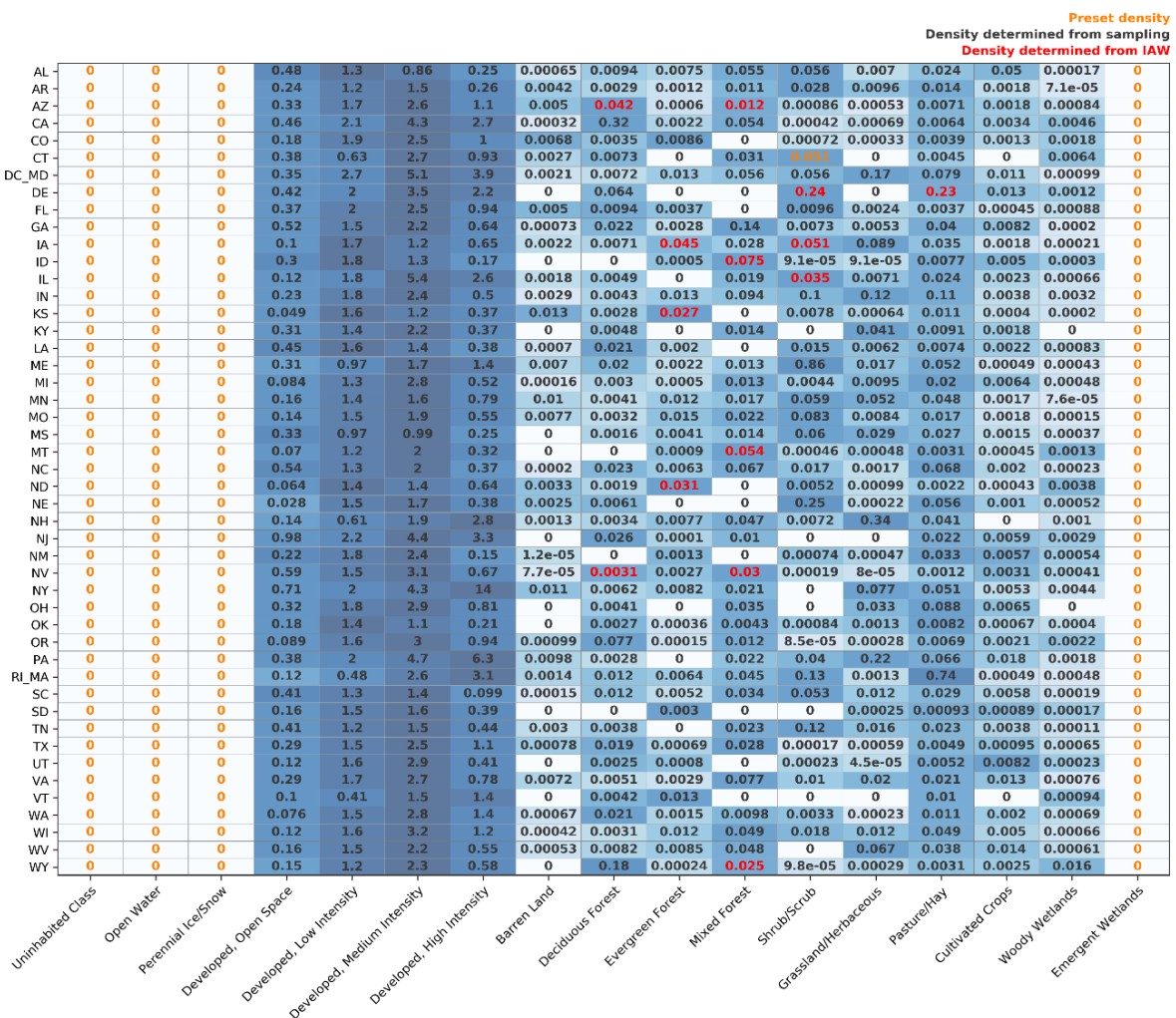

**Figure 2. Representative population densities determined using IDM. Note: The heatmap is scaled light blue to dark blue based on the sorted rank of densities for each state.**

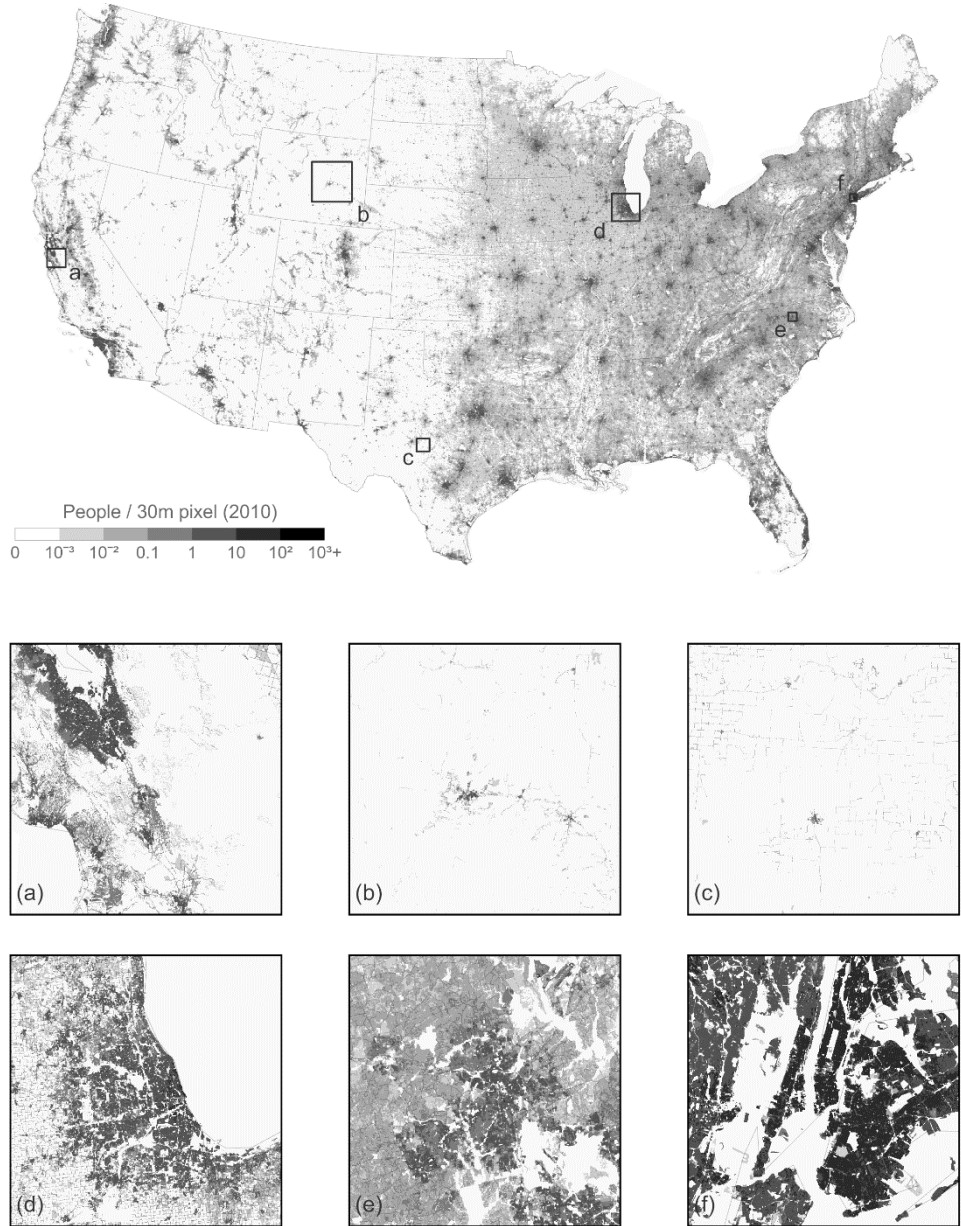

Figure 3. Population density estimated by Intelligent Dasymetric Mapping at 30 m spatial resolution for the conterminous United States and areas around (a) Santa Clara County, CA, (b) Natrona and Converse Counties, WY, (c) Concho County, TX, (d) metropolitan Chicago, IL, (e) Durham County, NC, and (f) metropolitan New York City, NY.

## 3.4  Assessment

The method we described above results in census block estimates equal to the census block numbers reported by the US Census Bureau; therefore, there is no cumulative error at the block level. To assess the validity and accuracy of our representative

population density estimates, we applied IDM on a larger source unit (i.e., census tract) using densities that we determined from the smaller source unit (i.e., census block). In other words, we disaggregated the recorded population for the census tract using block-level representative population densities (Fig. 4). We concatenated the state FIPS code, the county FIPS code, and the tract code to aggregate the census blocks by tract. The census population count for each tract was calculated by summing

the census population count from all the blocks inside each tract. An IDM population estimate for each block was then calculated by summing the per-pixel population densities estimated by using tracts as source units.

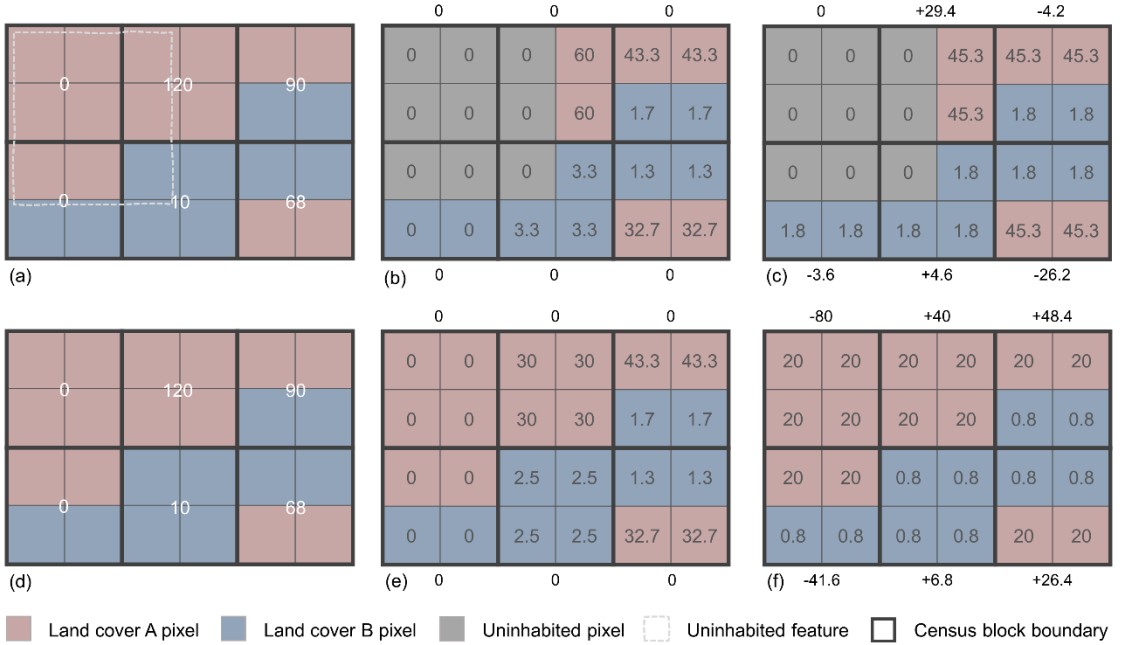

**Figure 4. A simulated illustration of six Census blocks and associated population within a single Census tract (a, d). This tract has two land cover types, A and B, with representative population densities estimated at 5.0 and 0.2 respectively and an uninhabited**
**feature that is new to the updated specification of uninhabited areas. Block level errors are provided adjacent to each block. Because our method has no cumulative error at the block level (b,e) we assessed our representative population densities by applying the densities at the tract level (i.e., no cumulative error at the tract level) with the updated specification of uninhabited areas (c) and the 2016 specification of uninhabited areas (f). In this illustration the tract has a MAE of 40.6 with the 2016 specification of uninhabited areas (f) and 11.3 with updated specification of uninhabited areas (c). Note: for illustrative purposes in this figure, we used the same**
**representative population density estimates for both updated (a-c) and 2016 (d-f) specifications. In practice the representative population density estimates for the updated and 2016 specifications were determined independently and most likely would have been different.**

Mean absolute error (MAE) and root mean square error (RMSE) were calculated to assess the error between the estimated block population and the recorded block population. RMSE was normalized by the mean block population within the summary

unit (i.e., state or county) to facilitate comparison between summary units (NRMSE). These metrics were calculated for each state and county in CONUS. Additionally, these metrics were calculated for the CONUS to compare model and zone performance and to facilitate comparison with other dasymetric population mapping efforts. The metrics were calculated as:

$$MAE_s = \frac{\sum_{b=1}^{n} |y_b - \hat{y}_b|}{n} \tag{7}$$

$$RMSE_s = \sqrt{\frac{\sum_{b=1}^{n} (y_b - \hat{y}_b)^2}{n}} \tag{8}$$

$$NRMSE_s = \frac{RMSE_s}{\overline{y}_s} \tag{9}$$

where:

$y_b$ = the census population count for block b

$\hat{y}_b$ = the estimated population for block b

$s$ = the unit for which census block errors are summarized. We used state , county, and CONUS.

$\overline{y}_s$ = the mean census block population count for unit $s$

$n$ = the number of blocks in unit $s$

We compared the RMSE and MAE between the 2016 specification of uninhabited areas and our updated specification by
running IDM for all CONUS states using both specifications. The 2016 specification of uninhabited areas used a preset density
of zero people/pixel for land cover classes open water, perennial ice/snow, and emergent herbaceous wetlands and included
areas with a slope of greater than 25%.

## 4    Results

### 4.1    **IDM Performance**

NRMSE ranged from 1.21 to 3.39 (Fig. 5;Table 2). The highest state NRMSE between census block population counts and
IDM estimated block population counts are for North Dakota with an RMSE that is 3.39 times the mean census block
population, Wyoming with an RMSE that is 2.91 times the mean census block population, and Montana with an RMSE that is
2.60 times the mean census block population (Table 2). The lowest NRMSE between census block population counts and IDM
estimated block population counts are for Connecticut with an RMSE that is 1.21 times the mean census block population,
Michigan with an RMSE that is 1.36 times the mean census block population, and New Jersey with an RMSE that is 1.38
times the mean census block population (Table 2). NRMSE was summarized by state and county (Fig. 6) highlighting areas
with highest (tending towards less densely populated) and lowest values (tending towards more urban).

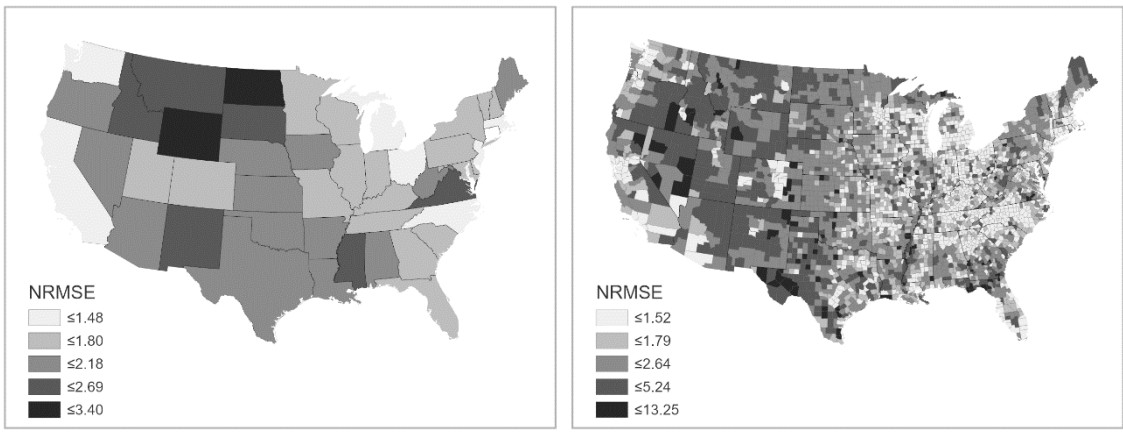

**Figure 5. NRMSE between block population estimates and block population census counts calculated for CONUS states (left) and counties (right). Block population was estimated by running IDM with census tracts as source units and applying representative population densities estimated by IDM using census blocks as preset densities.**

**Table 2. Census block average error by state after applying block level representative population densities for each ancillary class to census tracts and ensuring pycnophylactic integrity at the tract level. Change in error from the 2016 specification of uninhabited areas are in parenthesis.**

| State | RMSE | | NRMSE | | MAE | |
|-------|------|------|-------|------|-----|------|
| AL | 36.32 | (-1.48) | 1.86 | (-0.08) | 13.13 | (-0.6) |
| AR | 33.21 | (-0.92) | 2.07 | (-0.06) | 11.50 | (-0.35) |
| AZ | 52.18 | (-2.88) | 1.94 | (-0.11) | 15.84 | (-1.02) |
| CA | 78.32 | (-5.05) | 1.47 | (-0.09) | 28.31 | (-2.67) |
| CO | 44.50 | (-2.84) | 1.74 | (-0.11) | 15.46 | (-1.29) |
| CT | 65.43 | (-2.56) | 1.21 | (-0.05) | 28.29 | (-1.71) |
| DC_MD | 75.54 | (-3.13) | 1.73 | (-0.07) | 27.81 | (-1.71) |
| DE | 64.97 | (-4.45) | 1.71 | (-0.12) | 25.09 | (-1.97) |
| FL | 66.61 | (-3.29) | 1.68 | (-0.08) | 23.05 | (-1.33) |
| GA | 59.00 | (-4.44) | 1.75 | (-0.13) | 20.58 | (-1.65) |
| IA | 28.17 | (-0.99) | 1.97 | (-0.07) | 10.25 | (-0.45) |
| ID | 23.80 | (-1.46) | 2.23 | (-0.14) | 7.51 | (-0.46) |
| IL | 49.90 | (-3.37) | 1.72 | (-0.12) | 18.92 | (-1.55) |
| IN | 41.92 | (-1.74) | 1.69 | (-0.07) | 15.62 | (-0.86) |
| KS | 25.74 | (-0.99) | 2.11 | (-0.08) | 8.38 | (-0.34) |
| KY | 43.22 | (-3.17) | 1.57 | (-0.12) | 16.54 | (-1.17) |
| LA | 44.59 | (-1.22) | 1.96 | (-0.05) | 15.59 | (-0.58) |
| ME | 36.80 | (-1.56) | 1.88 | (-0.08) | 13.10 | (-0.48) |
| MI | 41.38 | (-2.47) | 1.36 | (-0.08) | 16.14 | (-1.29) |
| MN | 36.80 | (-1.87) | 1.77 | (-0.09) | 13.05 | (-0.83) |
| MO | 32.32 | (-1.28) | 1.80 | (-0.07) | 11.26 | (-0.63) |
| MS | 39.11 | (-0.98) | 2.21 | (-0.06) | 13.08 | (-0.58) |

| | | | | | | |
|---|---|---|---|---|---|---|
| **MT** | 19.66 | (-1.68) | 2.60 | (-0.22) | 6.08 | (-0.49) |
| **NC** | 48.84 | (-1.99) | 1.46 | (-0.06) | 19.83 | (-0.91) |
| **ND** | 17.27 | (-0.76) | 3.39 | (-0.15) | 4.61 | (-0.23) |
| **NE** | 21.04 | (-1.41) | 2.18 | (-0.15) | 6.91 | (-0.42) |
| **NH** | 46.90 | (-1.81) | 1.69 | (-0.07) | 17.33 | (-0.91) |
| **NJ** | 73.17 | (-7.85) | 1.38 | (-0.15) | 29.52 | (-3.72) |
| **NM** | 29.95 | (-1.96) | 2.41 | (-0.16) | 8.81 | (-0.65) |
| **NV** | 64.16 | (-2.33) | 1.98 | (-0.07) | 18.42 | (-1.07) |
| **NY** | 89.75 | (-5.56) | 1.60 | (-0.1) | 32.59 | (-2.89) |
| **OH** | 48.09 | (-2.6) | 1.48 | (-0.08) | 17.89 | (-1.42) |
| **OK** | 30.35 | (-1.29) | 2.12 | (-0.09) | 9.75 | (-0.49) |
| **OR** | 36.90 | (-2.57) | 1.87 | (-0.13) | 11.75 | (-0.88) |
| **PA** | 54.89 | (-3.99) | 1.79 | (-0.13) | 20.45 | (-1.67) |
| **RI_MA** | 63.00 | (-2.18) | 1.46 | (-0.05) | 25.41 | (-1.34) |
| **SC** | 44.46 | (-1.93) | 1.71 | (-0.07) | 16.53 | (-0.81) |
| **SD** | 23.18 | (-1.77) | 2.49 | (-0.19) | 7.52 | (-0.55) |
| **TN** | 44.51 | (-1.99) | 1.65 | (-0.07) | 16.63 | (-0.81) |
| **TX** | 58.50 | (-2.41) | 2.08 | (-0.09) | 18.47 | (-1.12) |
| **UT** | 41.00 | (-3.08) | 1.68 | (-0.13) | 13.38 | (-1.27) |
| **VA** | 63.94 | (-1.3) | 2.24 | (-0.05) | 17.45 | (-1.44) |
| **VT** | 34.48 | (-3.03) | 1.74 | (-0.15) | 11.83 | (-1.04) |
| **WA** | 51.45 | (-3.72) | 1.47 | (-0.11) | 18.99 | (-1.56) |
| **WI** | 36.56 | (-3.27) | 1.59 | (-0.14) | 13.61 | (-1.53) |
| **WV** | 27.07 | (-1.49) | 1.91 | (-0.11) | 9.58 | (-0.61) |
| **WY** | 19.35 | (-1.47) | 2.91 | (-0.22) | 5.79 | (-0.46) |

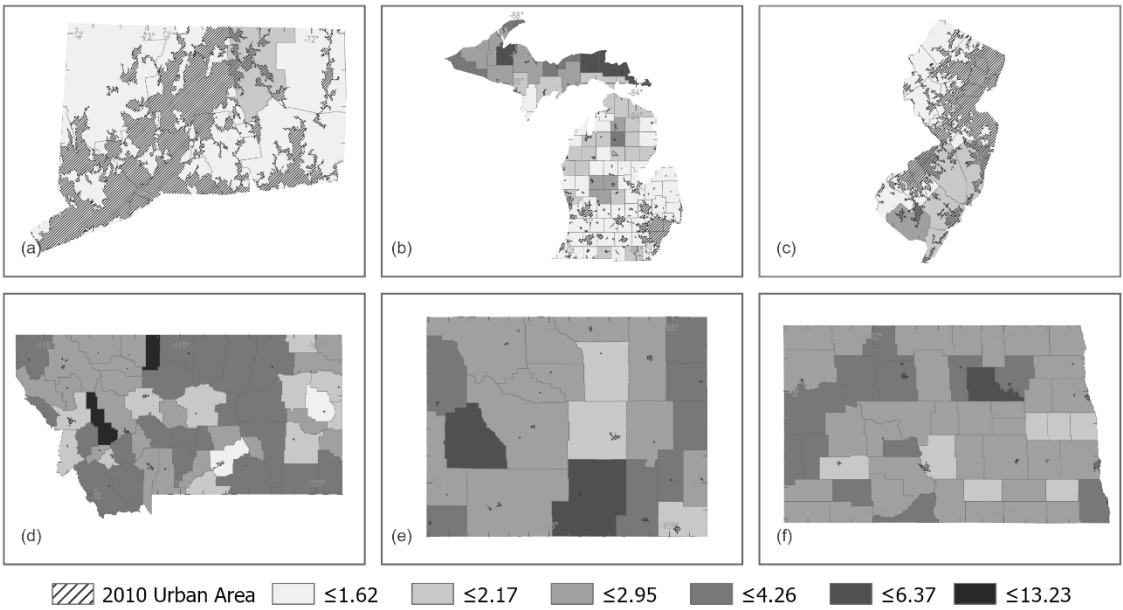

**Figure 6. Census urban areas and county NRMSE between census block population count and the estimated block population count from IDM for some of the states with the lowest NRMSE: (a) Connecticut, (b) Michigan, and (c) New Jersey and highest NRMSE: (d) Montana, (e) Wyoming, and (f) North Dakota.**

## 4.2 Uninhabited areas

The updated specification of uninhabited areas identified an additional 186,764,551 30 m pixels (~168,000 km2; an area slightly less than Washington State), as having zero population in comparison to the 2016 specification of uninhabited (Table 3). Recalling that the nature of IDM does not allow for population to be displaced beyond the original source unit (i.e., census block), our updated definition reallocated approximately 9.56 million people from uninhabited areas to areas that are more likely to be inhabited (Table 3; Fig. 7).

**Table 3. Count of pixels with and without population using the 2016 specification of uninhabited and the updated specification of uninhabited. Note. Counts include pixels within zero population blocks.**

|  | Pixels with population = 0 | Pixels with population > 0 | Population in Updated Uninhabited |
|---|---|---|---|
| 2016 Dasymetric Map | 4,338,376,834 | 4,641,477,058 | 9,564,807 |
| Updated Dasymetric Map | 4,525,111,385 | 4,454,742,507 | - |
| Difference | (186,734,551) | 186,734,551 | 9,564,807 |

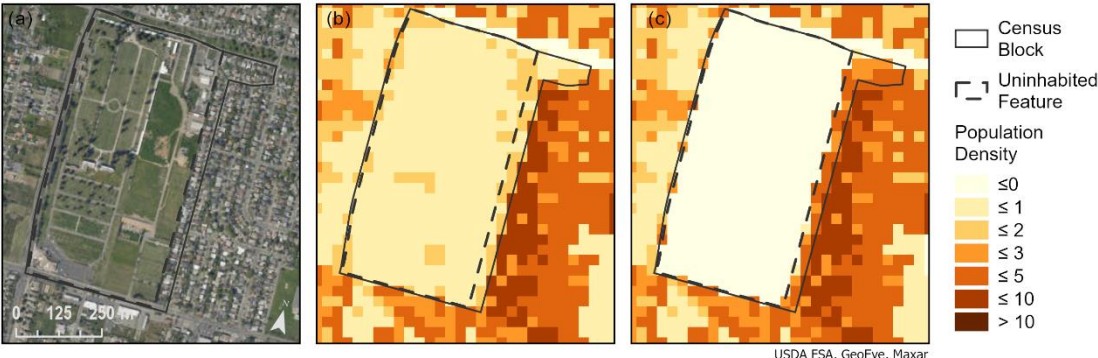

**Figure 7. A census block near Sacramento, California with a cemetery (i.e., uninhabited feature) covering most of the block and residential housing along the eastern border (a). IDM results with the 2016 specification of uninhabited (b) have population throughout the block while IDM results with the updated specification of uninhabited (c) have zero population for the cemetery and denser population along the eastern border.**

RMSE and MAE improved for all states with the expansion of uninhabited areas (Table 2). RMSE improved by an average of 2.46 persons per census block (σ = 1.37) and MAE improved by an average of 1.10 persons per census block (σ = 0.69) across all states. The most improved states were New Jersey and New York with a difference in RMSE of -7.85 and -5.56 and a difference in MAE of -3.72 and -2.89. Some of the least improved states were North Dakota and Arkansas with a difference in RMSE of -0.76 and -0.92 and a difference in MAE of -0.23 and -0.35 (Table 2). The expanded use of uninhabited areas improved overall RMSE for the CONUS when applying IDM both with nationally determined densities and with state level densities (Table 4).

**Table 4. Census block error for CONUS after applying block level representative population density for each ancillary class to census tracts and ensuring pycnophylactic integrity at the tract level.**

|  | RMSE | NRMSE | MAE |
|---|---|---|---|
| 2016 uninhabited areas (nationally determined densities) | 58.14 | 2.04 | 4.48 |
| Updated uninhabited areas (nationally determined densities) | 55.31 | 1.95 | 4.34 |
| 2016 uninhabited areas (state-by-state determined densities) | 54.80 | 1.93 | 4.30 |
| Updated uninhabited areas (state-by-state determined densities) | 51.93 | 1.83 | 4.15 |

## 5    Discussion

### 5.1 IDM Performance

IDM is a useful method to allocate population within heterogeneous source units. Intuitively, we would expect that identifying uninhabited areas within those source units would improve the accuracy of the allocation. Improvements in population model performance by adding variables for uninhabited areas were demonstrated by others (Fang and Jawitz, 2018). Many of the widely used models rely on multiple ancillary data layers to allocate population while acknowledging input data are often limited because of temporal constraints and necessity to cover large extents (Leyk et al., 2019). With a decrease in RMSE and

MAE for every CONUS state after identifying additional uninhabited areas, we have shown that with suitable, nationally consistent data improvements in population density estimates can be realized on regional, state, and country scales at a high spatial resolution.

Dasymetric mapping across an area as large and heterogenous as the CONUS benefited from the use of sub-national zones. Applying IDM on a state-by-state basis showed an improvement over using densities determined from a national analysis. A balance must be found in defining zones to ensure they are large enough to provide enough data to develop a useful model and small enough to maintain a suitable level of homogeneity within the zone. We attempted to further refine our product by using county-level 2013 United States Department of Agriculture Rural Urban Continuum Codes (RUCC) to create additional ancillary classes (U.S. Department of Agriculture, 2020). RUCC has nine classes but can be collapsed to official Office of Management and Budget metro and nonmetro county classification. We processed each state as described above but altered the ancillary raster so that the four NLCD developed classes within metro counties were given different values than the developed classes within nonmetro counties. This analysis did not show a significant difference in RMSE when compared to our state-by-state analysis. There may be a better scheme to highlight the differing population dynamics between rural and urban areas, but it would likely require more refined data than county-level. Dmowska and Stepinski (2017) used nationally determined densities but achieved lower error by taking advantage of a national land use map (Theobald, 2014) to identify uninhabited areas. We calculated a measure of error following the methods described in Dmowska and Stepinski and our error was comparable, but higher (43.17 mean block group RMSE versus 45.21 mean block group RMSE). However, the development of the Theobald land use map required a significant effort, and it is not clear whether those data will be available beyond 2010. The methods described here resulted in a comparable product using a variety of readily available and frequently updated data sources that can be appended as new sources become available or replaced entirely when more refined locally available data identifying uninhabited areas are available. The combination of identifying uninhabited areas along with the use of local or regional zones reduced RMSE and resulted in a more accurate dasymetric population product.

The representative population densities determined from IDM (Fig. 2) intuitively make sense. The four developed land cover classes were consistently orders of magnitude higher than all other land cover classes for all states. The densities were higher for "Developed, Low Intensity" compared to "Developed, Open Space" and were almost always higher for "Developed, Medium Intensity" compared to "Developed, Low Intensity". The "Developed, High Intensity" land cover class was, however, often lower than the medium intensity class likely due to the influence of highly developed and lightly populated industrial and commercial areas.

IDM's accuracy seems to be dependent on the spatial distribution of the population. States with the lowest NRMSE such as Connecticut and New Jersey tend to have larger urban areas with higher population counts well distributed throughout the state. This trend is likely from these states having a higher number of homogenous blocks from across the state identified as representative blocks. Conversely, states with the highest NRMSE such as North Dakota and Wyoming tend to be characterized

by small population centers surrounded by large sparsely populated lands (Fig. 6). These states tend to have fewer, less evenly distributed blocks eligible to be representative blocks. The same pattern seems to be repeated for counties. A given state's IDM representative population densities perform better in counties with a dispersed distribution of high population throughout the county rather than a stark difference between high population centers and surrounding sparsely populated areas. For example, some of the counties with the highest NRMSE in central and western Montana are characterized by low population blocks throughout the county with small concentrations of higher population blocks (Fig. 6). Furthermore, the counties in Michigan's upper peninsula with fewer urban areas tend to have higher NRMSE than the counties in the south with more distributed urban areas (Fig. 6).

## 5.2 Uncertainty and Limitations

Dasymetric modeling assumes a predictive relationship between ancillary data and a ground truth population surface, but, like any model, it only represents an approximation, with various sources of uncertainty. The core assumption is that population density is homogenous within ancillary classes, and many studies, including this one, put emphasis on refining those ancillary classes to make them more homogenous, or to allow for a degree of spatial autocorrelation in the heterogeneity of in-class density by using different estimates in different zones. As higher resolution ancillary data becomes more readily available, such efforts may face diminishing returns because a smaller spatial unit of measurement may have less sub-unit heterogeneity but more proportional uncertainty with regard to the population estimates (Azar et al., 2013;Nagle et al., 2014). Reducing uncertainty, therefore, is a matter of refining the fidelity between the ancillary data and the population density surface through a combination of automated and expert-guided techniques, often iteratively.

The decision to substitute the representative population density of shrub/scrub in Connecticut with a national average illustrates the importance of reviewing the output of IDM. Indeed, we would not expect shrub/scrub to be the most densely populated land cover class within Connecticut, and the estimated density is clearly an outlier when compared to other state's values for that same land cover class. While there are other values in our final estimates (Fig. 2) that may warrant additional attention, we believed this particular representative population density was so far outside the range of the other states we needed to consider an alternative value. It is imperative to review the results for logical consistency and consider modifications based on local knowledge before accepting the results.

Data for our uninhabited areas have a wide temporal range due to the varying frequency at which they are updated. For example, OSM data reflect the most recent edits made by contributers while the NLCD roads and energy development are from 2011. Although our population estimates are from the 2010 decennial census, the uninhabited areas are not restricted to 2010. There might be additional uninhabited areas since 2010. Furthermore, the rules applied to filter and refine uninhabited areas were determined for a national allocation of population. The EnviroAtlas IDM toolbox for ArcGIS Pro (https://github.com/USEPA/Dasymetric-Toolbox-ArcGISPro) or open source GIS (https://github.com/USEPA/Dasymetric-Toolbox-OpenSource) can be used to refine population estimates if more detailed local or regional data for uninhabited areas

are available. It is important to note that the accuracy of the population estimates is dependent on the accuracy of the input data. Some sources of uncertainty are the accuracy of the NLCD classification, the census block boundaries, and the boundaries and labels of various OSM, PAD-US, and NAVSTREETS layers.

## 5.3 Conclusion

In this study, we updated the existing dasymetric population map by EPA's EnviroAtlas by using additional geospatial datasets to expand the coverage of uninhabited areas. We used IDM developed by Mennis and Hultgren (2006) to estimate gridded 30 m population density for CONUS. The improved identification and masking of uninhabited areas improved the accuracy of population estimates for all CONUS states. Our accuracy assessment method showed that the IDM method was better at mirroring the Census block population counts of states with larger urban areas and smaller areas of sparsely populated land. The datasets and methods described here will be used to update the dasymetric population estimates for the CONUS once 2020 land cover and census data are available. Furthermore, the updated IDM toolbox will be used to specify uninhabited areas and to produce gridded population estimates for Alaska, Hawaii, Puerto Rico, and the Virgin Islands. The dasymetric population map and the IDM toolbox will be available in EnviroAtlas.

## 6 Code and Data Availability

The Dasymetric Toolbox for ArcGIS Pro (https://github.com/USEPA/Dasymetric-Toolbox-ArcGISPro) and Dasymetric Toolbox for Open Source GIS (https://github.com/USEPA/Dasymetric-Toolbox-OpenSource) are available on US EPA's GitHub page. The updated EnviroAtlas dasymetric population map at 30 m resolution for the CONUS is available via EPA's Environmental Dataset Gateway (Baynes et al., 2021; https://doi.org/10.23719/1522948). Data can also be accessed or viewed from EPA's EnviroAtlas (https://www.epa.gov/enviroatlas). Dasymetric population estimates for US States and Territories outside CONUS are in progress. Updates for all US States and Territories for the 2020 US Census are planned and will be available on EPA's EnviroAtlas.

## 7 Author contributions

JB and AN designed the study with input from TH. JB performed the analysis. All authors contributed to and approved the final manuscript.

## 8 Competing interests

The authors declare that they have no conflict of interest.

## 9 Acknowledgements

This paper has been reviewed in accordance with the U.S. Environmental Protection Agency's Center for Public Health and Environmental Assessment peer-review policies and approved for publication. Mention of trade names or commercial products does not constitute endorsement or recommendation for use. Statements in this publication reflect the authors' personal views and opinions and should not be construed to represent any determination of policy of the U.S. Environmental Protection Agency.

Maps throughout this article were created using ArcGIS® software by Esri. ArcGIS® and ArcMap™ are the intellectual property of Esri and are used herein under license. Copyright © Esri. All rights reserved. For more information about Esri® software, please visit www.esri.com. Use of OpenStreetMap data requires the following acknowledgment: "Map data copyrighted OpenStreetMap contributors and available from https://www.openstreetmap.org."

    We acknowledge Anam Khan for her expertise in developing the new IDM toolboxes and contributions to the manuscript
including her invaluable data analysis efforts. We acknowledge colleague Daniel Rosenbaum for providing the rail yards dataset along with James Wickham and Justin Bousquin for their comments on the manuscript. T. Hultgren's participation was underwritten by contract GS00Q09BGD0055 Task Order GSQ0017AJ0037 Mod 24 between US EPA and General Dynamics Information Technology, Inc.

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
