# Peer review of "Improving intelligent dasymetric mapping population density estimates at 30-meter resolution for the conterminous United States by excluding uninhabited areas"

_Earth System Science Data, 2021_

## Author Comment (AC1)

Based on the intelligent dasymetric mapping (IDM) method, the authors generated a set of high resolution population density product by using various datasets. This is interesting and significant since population change impacts almost every aspect of global change. However, it is noteworthy that the method described in this paper is not clear. To make it more persuasive, some suggestions includes:1) To better present how the data was processed and used, a flow chart is recommended; 2) It's better to list the spatial resolution for the raster data, and the feature types of vector data, such as point, line or polygon.

We thank the reviewer for their constructive comments and time spent with our paper. We agree with the reviewer's recommendations to clarify our methods. We updated Table 1 to include information on data types, resolution of input datasets, and clearer descriptions of IDM use. Additionally, we differentiated our vector (section 3.1) and raster (new section 3.2) pre-processing steps for identifying uninhabited areas and provided clearer descriptions for those steps. Finally, we have added Figure 1 to provide a better explanation of our workflow.

---

## Author Comment (AC2)

The authors describe the application of the IDM method to further improve the existing EPA EnviroAtlas Dasymetric Population Map for conterminous United States, mostly through the further expansion of uninhabitable areas. The resulting data product shows decreased estimation error in an assessment using census tract data as input and blocks as test units. The fine resolution of the gridded population density estimates will be very useful for various applications as described in the introduction of the manuscript.

We thank the reviewer for their constructive comments and time spent with our paper. We appreciate their acknowledged value of this product.

The result of defining more extensive and spatially expanded uninhabitable areas and using them for dasymetric refinement is very intuitive and expected. This strategy is about using robust constraining variables to reach estimates of higher accuracy.

This makes this study a rather incremental step to revise and improve an existing data product by including additional data layers that allow to identify more uninhabitable areas.

We agree that expanding the use of uninhabited areas to improve population density estimates is intuitive; however, the results were not guaranteed. We were pleased to see improvement across every state in the nation. We also made two subtle, yet important, adjustments to the IDM method in Eq. 3 and Eq. 5. Those adjustments, included in the newly developed IDM toolboxes, and these results give us confidence to use similar methods to further refine population estimates as needed in the future.

IDM is a slightly dated (though still effective) method for population refinement. Given the nature of the data and the statistical frameworks available it is surprising that the authors have not tested different methods (e.g., machine learning) to cross-compare performance and resulting data quality. The given problem at hand seems to be one that could be treated effectively with learning frameworks.

The authors agree that machine learning methods have the potential to solve the limitation of assuming a single weight across a heterogenous class and ultimately improving results. We are not yet confident the same techniques and hyperparameters applied at a large extent would be suitable at a local or regional extent. Our goals were two-fold; to produce an improved estimate of population density across the nation and to provide others the means to use those same methods for their own selected areas. IDM has several attractive qualities for these goals. It provides reasonable estimates of population density from a minimal number of required inputs, an ancillary raster and source units, and has a relatively direct workflow. Machine learning may improve results but at a cost of complicating the method and potentially prohibiting its use at various scales and by broad audiences. To address this comment in the manuscript, we have better explained our rationale for selecting IDM in the introduction.

Estimating the average population density of target zones across a whole state is not effective and represents a well-known limitation of IDM. The authors mention the further refinement of their estimation procedure for urban and rural classes as future work. However, it seems that this would have been an important step already in this data product revision. The estimation of population densities for different urban classes (intensity of development) will result in much more accurate data layers. This would allow the model to proceed at the scale at which variation is observed across counties and states.

We thank the reviewer for this suggestion. To answer this lingering question from our paper, we followed our methods while differentiating developed classes in metro versus nonmetro counties. Four new classes were added to the ancillary raster for each state: developed – open space in metro, developed – low intensity in metro, developed – medium intensity in metro, and developed – high intensity in metro. Metro areas were identified using county-level 2013 Rural-Urban Continuum Codes from the United

States Department of Agriculture. We found no meaningful difference in accuracy between this method and our existing method. We described this additional analysis in our discussion, and we have removed this suggestion for refinement from our paper. We left open the possibility that more spatially refined data identifying the urban rural continuum beyond county level classification could improve accuracy. We also provided an explanation of our selection of states for zones in section 3.4. Finally, we calculated RMSE for the CONUS to measure the effect of state-level zones and facilitate comparisons with other zoning schemes in the future. We included those results in a new Table 4.

General: The different data layers and processing steps need more precise descriptions.
We agree the data layers and processing steps could benefit from more precise descriptions. We updated Table 1 to include information on data types, resolution of input datasets, and clearer descriptions of IDM use. Additionally, we differentiated our vector (section 3.1) and raster (new section 3.2) pre-processing steps for identifying uninhabited areas and provided clearer descriptions for those steps. Finally, we have added Figure 1 to provide a better explanation of our workflow.

The authors are also encouraged to provide a broader background on dasymetric modeling including uncertainty concerns and typical limitations reported in the literature.

We agree a broader background on dasymetric modeling uncertainty and limitations is important. We have added content to section 5.2 (Uncertainty and Limitations) addressing this.

Also, there are other fine-resolution population datasets, and it is recommended that the authors make sure their data layers are compared to those existing products.

We agree there are other products developed with similar methods and our paper could benefit from comparison. Based on another comment, we have referred to the national dasymetric population map described in Dmowska and Stepinski (2017) and compared our results with that effort in our discussion. Furthermore, we have added Table 4 that provides error statistics for our product at a national scale to facilitate comparisons with other CONUS dasymetric mapping efforts.

---

## Author Comment (AC3)

The paper describes 30m resolution population density maps for the US. The resultant map is a product of disaggregating 2010 US Census block-level data using an NLCD and additional geospatial layers to identify uninhabited areas. The paper describes only EPA EnviroAtlas Dasymetric Population maps and does not mention other high-resolution population maps available for the entire conterminous US. For example, a 30m resolution population density maps for the conterminous US has been already created and it is available via the SocScape project (http://socscape.edu.pl, see paper for further information https://www.sciencedirect.com/science/article/pii/S0198971516301983). The 2010 map available via the SocScape project is a product of disaggregating block-level population data using NLCD2011 and land use map for further identification of uninhabited areas (Theobald 2014) and uses a similar approach to dasymetric modeling (Mennis and Hultgren, 2006).

Authors should describe other high-resolution maps available for the US and discuss how their product advances the map that already exists (available via the SocScape project)

We thank Dr. Dmowska for her comments and introducing us to the SocScape project. We are intrigued by both your population grids and racial diversity grids. The primary focus of our paper was to advance the intelligent dasymetric mapping (IDM) method specifically. To that end, we developed two modern IDM toolboxes with new functionality to mask uninhabited areas, clarified Eq. 3, modified Eq. 5, and released an updated national 30m dasymetric population map for the CONUS. We initially chose to limit our comparison to the 2016 version of the EnviroAtlas 30 m dasymetric population map because that map was developed with the specific methods we were intending to improve.

The authors agree the dasymetric methods are similar, particularly for ancillary class densities determined with sampling, but the identification of uninhabited areas and our use of sub-national zones are different. In a comparison of national level RMSE using the methods described in Dmowska and Stepinski (2017) our results were comparable, with our RMSE being slightly higher. The Theobald land use map (Theobald 2014) is likely a better proxy for identifying uninhabited areas for CONUS in 2010. However, the Theobold land use map was developed using 2006 land cover data and we question what alternatives are available if the Theobald map is not updated for 2020 or for areas that do not have the benefit of comprehensive land use data. The combination of identifying uninhabited areas along with the use of regional zones was beneficial for our product and would likely be useful for others.

The authors acknowledge that the SocScape-30 population density map described in Dmowska and Stepinski (2017) is important and warrants comparison. We particularly appreciate Fig. 3 as it effectively illustrates the challenges of assuming homogenous density of ancillary classes. We referenced Dmowska and Stepinski (2017) in our introduction and added a comparison to the project in our discussion.